# Congenital Anomalies in American Crocodile (*Crocodylus acutus*, Cuvier, 1807) Embryos from a Farm Breeder in Colombia

**DOI:** 10.3390/vetsci11070317

**Published:** 2024-07-15

**Authors:** Oscar Sierra Serrano, Andreia Garcês, Isabel Pires, John Alexander Calderón Mateus, Juan Medina Olivera, Jhesteiner Julio Dávila

**Affiliations:** 1Independent Researcher, Sincelejo 700001, Colombia; candelilla.azul@gmail.com; 2Wildlife Rehabilitation Center and Exotic Service, Teaching Veterinary Hospital University of Trás-os-Montes and Alto Douro, Quinta dos Prados, 5000-801 Vila Real, Portugal; 3Centre for Animal Sciences and Veterinary Studies, Associate Laboratory for Animal and Veterinary Science—AL4AnimalS, University of Trás-os-Montes e Alto Douro, Quinta dos Prados, 5000-801 Vila Real, Portugal; ipires@utad.pt; 4Parque Vivarium del Caribe, Via-Pontezuela-Bayunca, Cartagena 130001, Colombia; jhoncalderon@gmail.com; 5Cocodrilos de Colombia, Cartagena 130001, Colombia; 6University of Córdoba, Monteria, Poniente Sur, Córdoba 14071, Colombia; jmedinaolivera@correo.uncordoba.edu.co; 7Instituto de Investigaciones Geográficas de Investigaciones Geográficas y Ambientales del Caribe (GEOCARIBE), Poniente Sur, Córbora 14071, Colombia; 8Grupo de Investigación Biología Evolutiva, Programa de Biología, Facultad de Educación y Ciencias, Universidad de Sucre, Sincelejo, 5-267 Barrio Puerta Roja, Sincelejo 700001, Colombia; jhesteiner13@gmail.com

**Keywords:** teratology, *Crocodylus acutus*, caimán aguja, embryos, malformations

## Abstract

**Simple Summary:**

Congenital defects have been described in almost every vertebrate group. In crocodiles, teratology alterations have been described in captive animals (pets, zoos, farms) such as *Crocodylus niloticus* and *Gavialis gangeticus*. The present study aimed to characterize congenital malformations of *C. acutus* from a farm in Lomas de Matunilla, Ballestas, Bolívar, Colombia. The analyzed eggs presented macroscopic malformations, with 42 different types of anomalies observed. Limb and tail malformations (29%) were the most common changes observed.

**Abstract:**

The American crocodile (*Crocodylus acutus*, Cuvier, 1807) (Class Reptilia, Family Crocodylidae) is a crocodile species inhabiting the Neotropics. Congenital defects have been described in almost every vertebrate group. In crocodiles, teratology alterations have been described in captive animals (pets, zoos, farms) such as *Crocodylus niloticus* or *Gavialis gangeticus*. The present study aimed to characterize congenital malformations of *C. acutus* from a farm in Lomas de Matunilla, Ballestas, Bolívar, Colombia. A total of 550 unhatched eggs were examined after embryo death. A total of 61 embryos presented malformations, with 42 different types of anomalies observed. Limb and tail malformations (29%) were the most common malformations observed. Several malformations, such as cephalothoracopagus, thoracopagus, sternopagus, xiphopagus twins, campylorrachis scoliosa, and acrania, were documented in crocodiles for the first time. Research in teratology enhances our understanding of crocodile biology. It plays a role in their conservation and management, thus helping to ensure the long-term viability of these species in their natural habitats.

## 1. Introduction

American crocodile (*Crocodylus acutus*) (Class Reptilia, Family Crocodylidae) is a species of crocodile that inhabits the Neotropics like Belize, Colombia, Costa Rica, Cuba, the Dominican Republic, Ecuador, El Salvador, Guatemala, Haiti, Honduras, Jamaica, Mexico, Nicaragua, Panama, Peru, the United States (Florida), and Venezuela [1,2]. According to the International Union for Conservation of Nature (IUCN)—Red List, this species is considered Vulnerable [3]. Its habitat consists mainly of coastal areas [4]. *C. acutus* breed in late fall or early winter, and in February or March, the females lay 30 to 70 eggs in nests of sand, mud, and dead vegetation along the water bodies [5]. The incubation period is about 75–80 days. The temperature at which the eggs are incubated (dependent on the environment) can influence the sex of the hatchlings, a phenomenon known as temperature-dependent sex determination (TSD). Warm temperatures during incubation tend to produce females, while cooler temperatures produce males [6]. Alteration of the temperatures can alter the embryos [7].

Like other species, *C. acutus* faces various threats [3,8]. Expanding cities and towns into coastal and wetland areas leads to the loss of crucial nesting and basking sites. The conversion of wetlands and coastal regions into agricultural lands destroys crocodile habitats. Agricultural runoff, industrial waste, and urban pollution degrade the quality of water bodies where these animals live. Dam construction and water diversion for agriculture and urban use can alter the natural flow of rivers and streams [9]. In addition, illegal hunting exists due to the demand for crocodile skins, leather products, and meat [10]. Climate change leads to rising sea levels inundating nesting sites, and temperature changes can skew sex ratios, impacting population dynamics [11]. With the rise of human–crocodile conflicts, these animals sometimes prey on livestock or come to attack humans, leading to retaliatory killings [12].

Congenital defects have been described in almost every vertebrate group [13,14,15], but in wild animals, the descriptions of these anomalies are scarce, and the information is virtually non-existent [16]. They are more commonly observed in captive animals [17]. Most of the individuals who carry these malformations die before birth. If they survive, most malformations are only detected during a post-mortem exam [18]. Teratogens, agents, or factors that can disturb the normal development of an embryo can include environmental agents like drugs, infectious diseases (e.g., viruses, bacteria, and fungi), radiation, temperature, pollution (e.g., heavy metals, and pesticides), or certain maternal conditions [13,18,19].

In reptiles, a wide variety of congenital anomalies has been reported in several species [20]. For example, Martín-del-Campo et al. (2021) [21] showed the presence of malformation in three species of sea turtles associated with different environmental factors such as the incubation temperature, humidity, and the status of feeding areas [21]. Other studies in South American pit vipers (*Bothrops jararaca*) and South American rattlesnakes (*Crotalus durissus*) linked the malformation to the presence of chemical compounds such as pesticides and herbicides [16].

In crocodiles, teratology alterations have been described in captive animals (pets, zoos, farms) in species such as *Crocodylus niloticus*, *Gavialis gangeticus*, *Osteolaemus tetraspis*, *Alligator mississippiensis*, *C. porosus*, *C. johnsoni*, *C. palustris*, or *C. moreletii* [22,23,24]. Not all malformations can be considered as being from congenital origin, with some resulting from the exposure of embryos to environmental stressors [18]. Pollution is a major concern, particularly chemical contaminants such as pesticides, heavy metals, and endocrine-disrupting compounds that can accumulate in aquatic ecosystems [25]. These substances can interfere with normal developmental processes, leading to physical abnormalities. Habitat degradation due to urbanization, agriculture, and industrial activities can also play a role. Destruction of nesting sites and changes in water quality can create stressful conditions that increase the likelihood of malformations. Climate change and associated effects, like temperature fluctuations and altered precipitation patterns, can also disrupt embryonic development since crocodile sex determination and growth rates are temperature-dependent [22,26].

The present study aimed to characterize congenital malformations observed in American crocodiles (*Crocodylus acutus*) from a farm in Colombia.

## 2. Materials and Methods

Eggs of the American crocodile (*Crocodylus acutus* Cuvier, 1807) from a farm located in Lomas de Matunilla, Ballestas, Bolívar, Colombia (10°10′06″ N 75°28′46″ W) (Figure 1), were analyzed during Colombia’s dry season (December to March) in 2023.

From 61 nests across 15 lakes, 1773 eggs were laid from 61 females. The eggs were incubated and hatched from 25 April 2023 to 31 May 2023. Of the 1773 eggs, 1242 were incubated, with 48 eggs broken and 470 infertile eggs. Table 1 shows the information regarding nests, lakes, offspring, collection date, incubated eggs, laid eggs, broken eggs, infertile eggs, collected eggs, and dead embryos.

All the eggs that did not hatch in the expected period (75–80 days) were analyzed. After embryo death, stillborn crocodiles were removed from the eggs and measured, systematically examined, and photographed with a scale. All the gross malformations were recorded in an Excel file and then categorized by different anatomical regions into skin, spine, head (excluding eye and nose), nose, limbs, tail, and other malformations.

## 3. Results

From the 1773 eggs laid, 550 eggs with dead embryos were analyzed from those 61 embryos that presented malformations. Overall, 95.7% (*n* = 57) presented more than one malformation. Forty-two different types of anomalies were observed (Table 2).

Regarding body regions, 4.9% (10/202) affected the skin, 9.8% (20/202) the spine, 29% (61/204) the limbs and tail, and 25% (51/202) the head. Other malformations were 27.9% (Figure 2). The main malformation by region observed were as described below:

### 3.1. Skin Malformations

*Ephitheliogenesis imperfecta* (a localized region of abnormal skin) was observed in five animals, two of which had incomplete scale formation (Figure 3). Alterations of scale color were leucism (*n* = 2), where the scales were white crocodiles with dark markings and regional depigmentation to white (*n* = 3) (Figure 4).

### 3.2. Spine Malformations

The main spine malformations were observed in fourteen animals with scoliosis (lateral curvature of the spinal column), three with lordosis (ventrodorsal deviation, curvature of the spinal column with a ventral convexity) (Figure 5), two with kyphosis (Increased dorsal convexity in the curvature of the spinal column as viewed from the side) (Figure 6), and one with campylorrachis scoliosa (curvature of the spine a lack of vertebrae and spinal cord caudal to the thoracic region) (Figure 7).

### 3.3. Head

The main malformations observed in the head were one anencephaly/prosencephalic hypoplasia (absence of the cranial region of the head, with the brain absent or reduced), twelve laterognathia (mandible pointing up and jaw pointing down), three meningoencephalocete (herniation of the brain and meninges through a cranial opening covered by skin), one brachygnathia (short lower mandible), four microcephaly (small head), three maxillary micrognathia (short upper jaw), one external hydrocephalus (abnormal accumulation of cerebrospinal fluid within the ventricles and/or subarachnoid spaces, increase in head circumference, combined with enlarged subarachnoid spaces), seven maxillary macrognathia (longer low jaw), three acrania (absence skull), one acephaslostomia, one maxilla agnathia (absent upper jaw), and one brachycephalic skull (shape of a skull shorter than average in its species) (Figure 8, Figure 9, Figure 10, Figure 11, Figure 12 and Figure 13).

### 3.4. Limbs and Tail Malformations

In the limbs, both anterior and posterior, observed micromeli (short member) (*n* = 1), polydactyly (extra digit) (*n* = 2) (Figure 14), carpal flexure (*n* = 1), malroted members (*n* = 1) and arthrogryposis (*n* = 1).

Alterations of the tail observed were curled tail (curved into nearly a full circle) (*n* = 17), kinked tail (localized undulation(s) of the tail) (*n* = 18), acaudia (absent tail) (*n* = 16), brachyuri (*n* = 3), and bent tail (*n* = 2) (Figure 15).

### 3.5. Other Malformations

In Achondroplasia (dwarfism), the long bones are abnormally short, although the trunk is of normal length and the abdomen is large (round)—the size of normal individuals. The head is moderately enlarged and flattened. This malformation was observed in four individuals (Figure 16). Four cases of twins were observed. Cephalothoracopagus (Figure 17) were symmetrical monozygotic twin crocodiles that conjoined from the head down to the thorax with one head, one face, one neck, a single thorax, two abdomen, two umbilical cords, eight limbs, and two tails. There were three monozygotic symmetrical twins (identical twins) (Figure 18) and one thoracopagus, sternopagus, and xiphopagus twin (Twins attached by the thorax, sternum, and xiphoid process). We also observed one case of a distended celomic cavity, one case of Ectopia cordis (externalized heart), one case of gastroschisis (externalization of the internal organs abdomen), and 47 cases of yolk sac retention.

## 4. Discussion

In this work, the authors describe the congenital deformities in American crocodiles (*Crocodylus acutus*). To the the authors’ knowledge, the present study is the first of this kind. According to studies performed [27], crocodiles have a low prevalence of deformities even in captivity. In *Gavialis gangeticus*, deformities were detected in 6% of the 1061 hatchlings examined [28].

In this study, all the animals were born in captivity and died during gestation, except for one individual. This animal was born without a tail; it would be difficult for it to survive in the wild since it could not swim to the surface [24], but it is more likely to survive in captivity. Limbs and tail malformations (29%) were the most common changes observed, followed by head (25%). These results are similar to what has been described by other authors [6,24,29]. Almost every congenital malformation observed in this study has been reported in other species of crocodiles [23,24,26]. In the cases described were observed malformations such as cephalothoracopagus, thoracopagus, sternopagus, xiphopagus twins, campylorrachis scoliosa, acephaslostomia, and acrania, that were not described in crocodiles to the moment.

Malformations in wild animals are rarely described [22,27]. These congenital malformations can be associated with many factors, such as hatchlings from very young and very old females, genetic causes, malnutrition of the parents, defective incubation, and carcinogenic agents [30].

For example, *Alligator mississippiensis* from Lake Apopka (USA) embryos presented a large number of malformations associated with a chemical spill of sulfuric acid, dicofol, organochlorine pesticides, and Polychlorinated biphenyls (PCBs) [29,31]. South American pit vipers (*Bothrops jararaca*) and South American rattlesnakes (*Crotalus durissus*) malformations have been linked to the increased use of pesticides and herbicides in the agriculture fields near their habitat [16]. Benzo(a)pyrene and dibenzo(a)anthracene, polychlorinated biphenyls (PCBs), and heavy metals (As, Cu, Cd, Hg, Pb) were related to the development of malformations in the semi-aquatic turtles such as *Chelydra serpentina* and *Chrysemy spicta* from the John Heinz National Wildlife Refuge (USA) [32].

There is the possibility that genes can be associated with these malformations if both parents are carriers. In gharial populations from Nepal, a ‘blind’ gene has been suspected to occur [28]. In spite of our study, we did not confirm the etiology of the congenital malformations; a combination of many factors, such as the females’ age, inbreeding, incubation errors, and chemical pollutants in the water (the farms’ water originates in very polluted rivers in Columbia) should be considered. Although no samples were collected from the water in the farm lakes, it is very possible that pesticides like Organochlorine and heavy metals are present in such water since these compounds have been detected in other rivers in Colombia [29].

Although other studies have reported the occurrence of malformations in crocodiles and other reptiles, there is still a large gap concerning the classification of these malformations and defining the frequency with which they occur. Crocodiles are ectothermic carnivores that live in aquatic environments and may act as sentinel species [33,34]. In the future, more detailed studies should be performed to find the potential role of genetic and environmental factors in the occurrence of the reported malformations in these species.

## 5. Conclusions

This study describes congenital deformities in American crocodiles (*Crocodylus acutus*). It is important to note that research in teratology not only enhances our understanding of the biology of crocodiles but also plays a role in their conservation and management, helping to ensure the long-term viability of these species in their natural habitats. Identifying the causes of developmental abnormalities can contribute to conservation efforts by addressing potential threats and improving management strategies. Although other studies have reported the occurrence of malformations in crocodiles, there is still a large gap concerning the classification of these malformations and the frequency at which they occur. Examining abnormal development can also offer insights into the genetic diversity of crocodile populations and potential genetic stressors.

## Figures and Tables

**Figure 1 vetsci-11-00317-f001:**
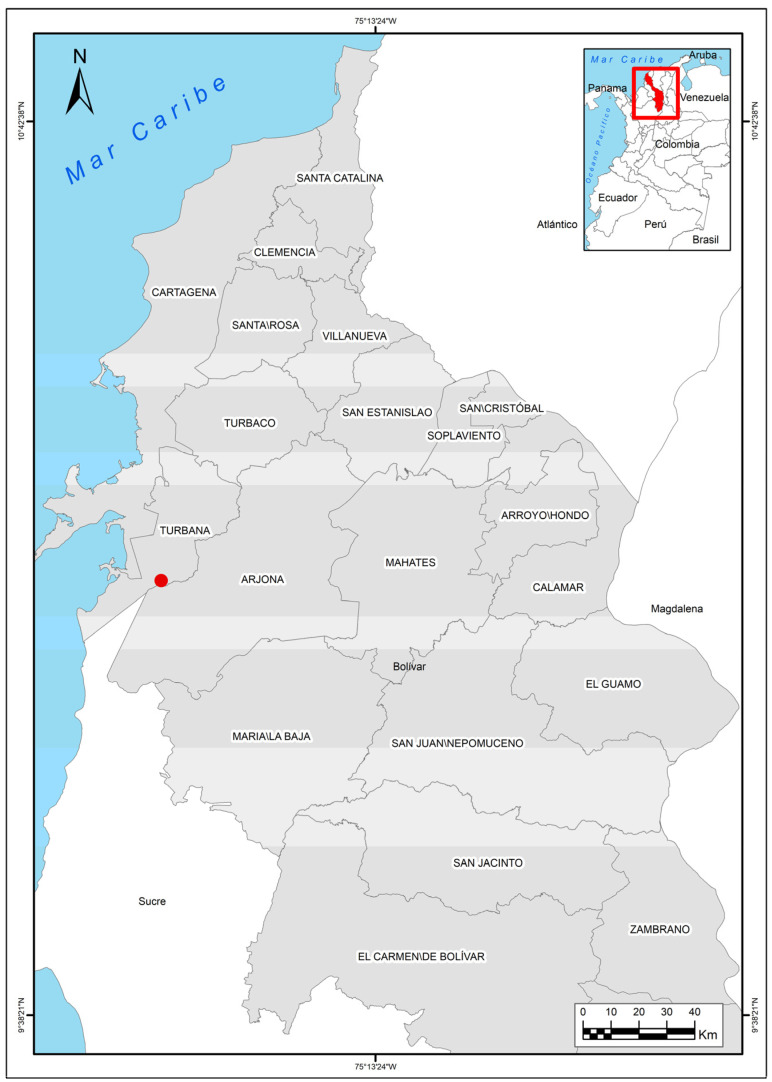
Colombia map with the location of the American crocodile (*Crocodylus acutus* Cuvier, 1807) farm.

**Figure 2 vetsci-11-00317-f002:**
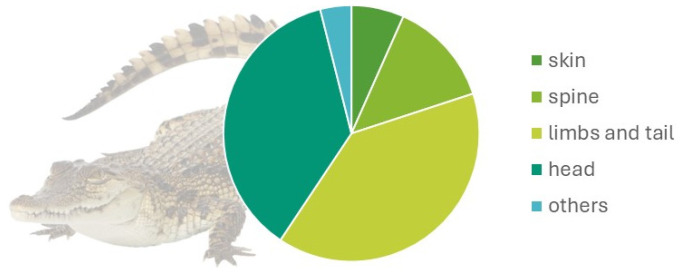
Malformation distribution by anatomical region in 61 embryos of the American crocodile (*Crocodylus acutus* Cuvier, 1807).

**Figure 3 vetsci-11-00317-f003:**
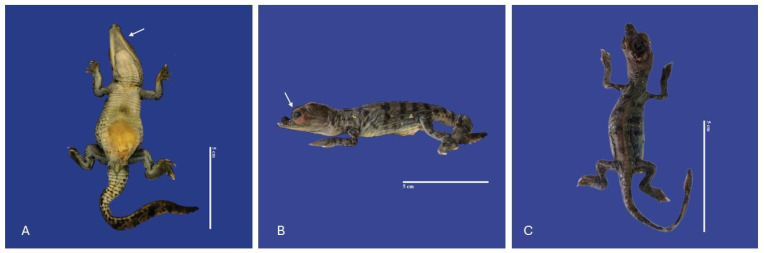
Ephitheliogenesis imperfecta in the lower jaw (↑) (**A**), eyes (↑) (**B**), absence of scales on the entire body (**C**) in *Crocodylus acutus* embryos. Scale bar = 5 cm.

**Figure 4 vetsci-11-00317-f004:**
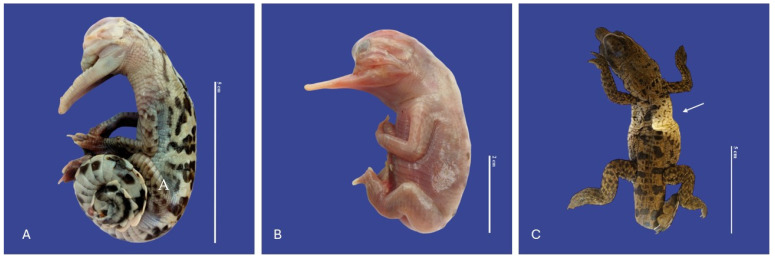
Leucism (**A**,**B**), depigmentation (↑) (**C**) *Crocodylus acutus* embryos. Scale bar = 2 cm and 5 cm.

**Figure 5 vetsci-11-00317-f005:**
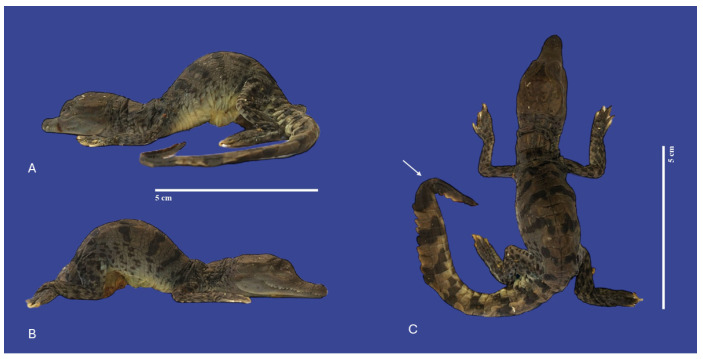
Lordosis, kinked tail (↑), and yolk sac retention with non-closure of the abdominal wall in a *Crocodylus acutus* embryo ((**A**,**B**)—Lateral view, (**C**)—dorsoventral view). Scale bar = 5 cm.

**Figure 6 vetsci-11-00317-f006:**
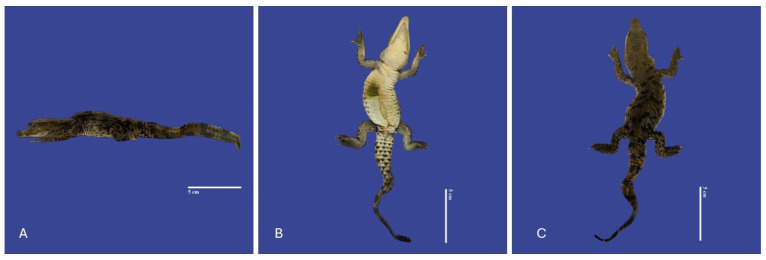
Yolk sac retention with non-closure of the abdominal wall, kinked tail, scoliosis and kyphosis in a *Crocodylus acutus* embryo. ((**A**)—Lateral view, (**B**)—ventral view, (**C**)—dorsoventral view). Scale bar = 5 cm.

**Figure 7 vetsci-11-00317-f007:**
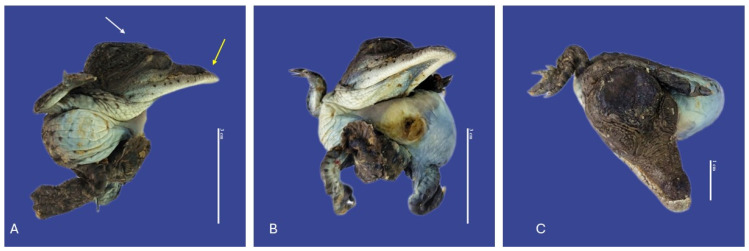
Yolk sac retention with non-closure of the abdominal wall, maxillary macrognathia (↑ yellow), acrania (↑ white), curved tail, and campylorrachis scoliosa in *Crocodylus acutus* embryo. (**A**,**B**)—Lateral view, (**C**)—cranial view). Scale bar = 3 cm and 1 cm.

**Figure 8 vetsci-11-00317-f008:**
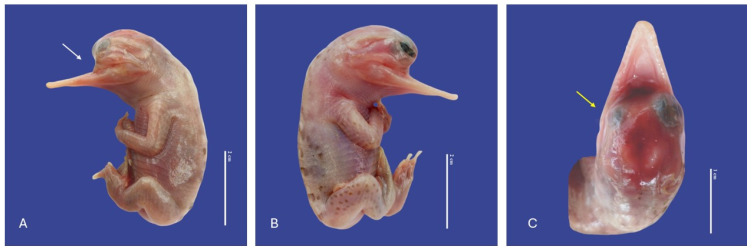
Maxilla agnathia (↑ white), microphthalmia (↑ yellow), acaudia, and leucism in *Crocodylus acutus* embryo. ((**A**,**B**)—Lateral view, (**C**)—cranial view). Scale bar = 2 cm and 1 cm.

**Figure 9 vetsci-11-00317-f009:**
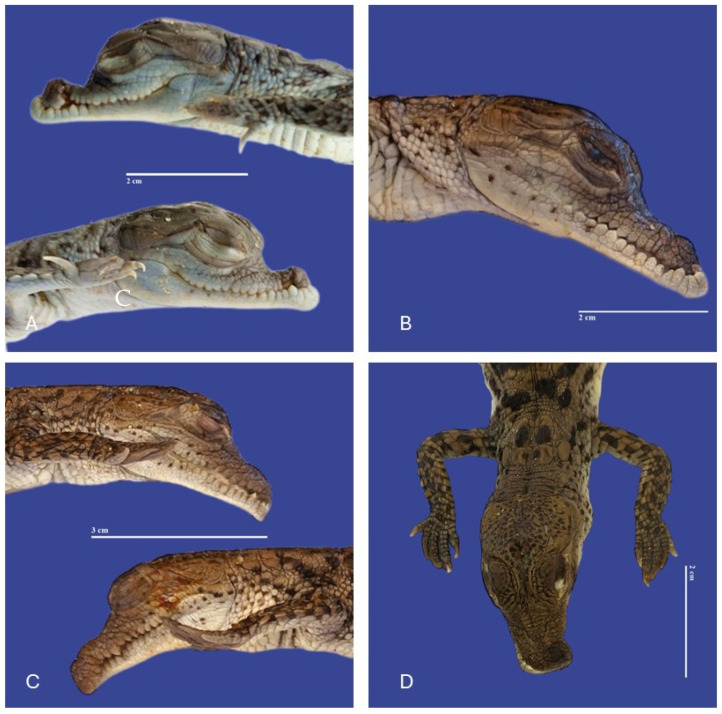
(**A**) Maxillary micrognathia and atresia; (**B**) maxillary micrognathia; (**C**) maxillary macrognathia and laterognathia; (**D**) Laterognathia in *Crocodylus acutus* embryos. Scale bar = 2 cm and 3 cm.

**Figure 10 vetsci-11-00317-f010:**
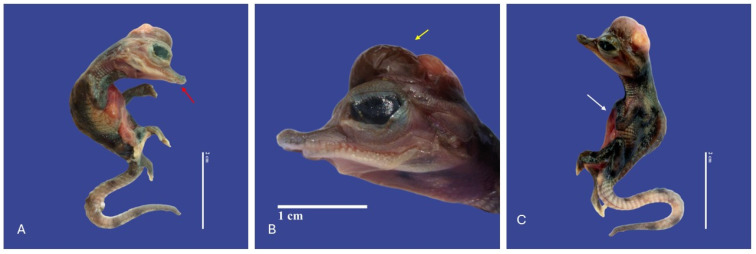
Depigmentation of the skin, mandibular micrognathia (↑ red), microcephaly, gastroschisis, ectopia cordis (↑ white), schistosomia, kyphosis, meningoencephalocele (↑ yellow), bend tail, epitheliogenesis imperfecta in all the body in *Crocodylus acutus* embryo ((**A**,**C**)—Lateral view, (**B**)—Head). Scale bar = 2 cm and 1 cm.

**Figure 11 vetsci-11-00317-f011:**
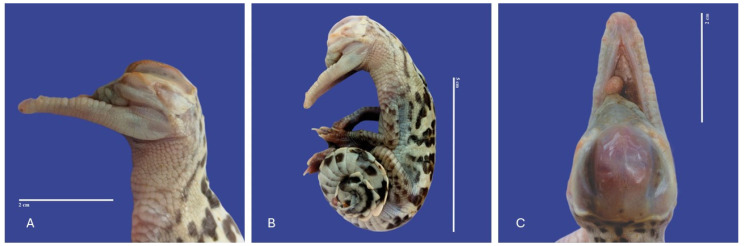
Leucism, acephalostomia, anophthalmia, atresia, and curly tail in *Crocodylus acutus* embryo. ((**A**)—Lateral head view, (**B**)—Lateral body view, (**C**)—cranial view). Scale bar = 2 cm and 5 cm.

**Figure 12 vetsci-11-00317-f012:**
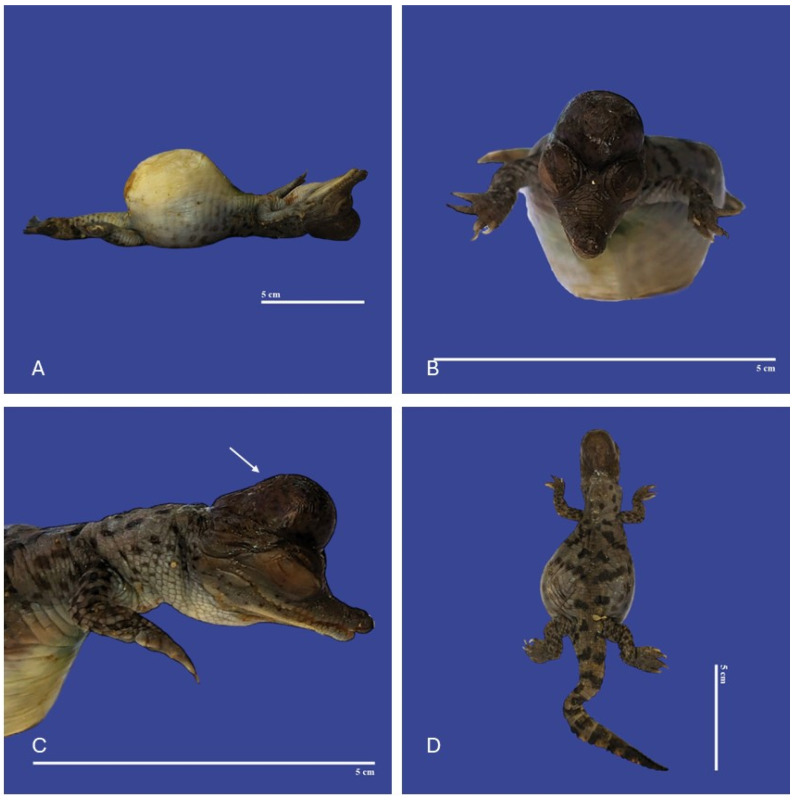
Yolk sac retention, meningoencephalocele (↑ white), kinked tail, exophthalmia, and brachygnathia in *Crocodylus acutus* embryo. ((**A**)—Lateral body view, (**B**)—Cranial view, (**C**)—Lateral head view, (**D**)—dorsoventral view). Scale bar = 5 cm.

**Figure 13 vetsci-11-00317-f013:**
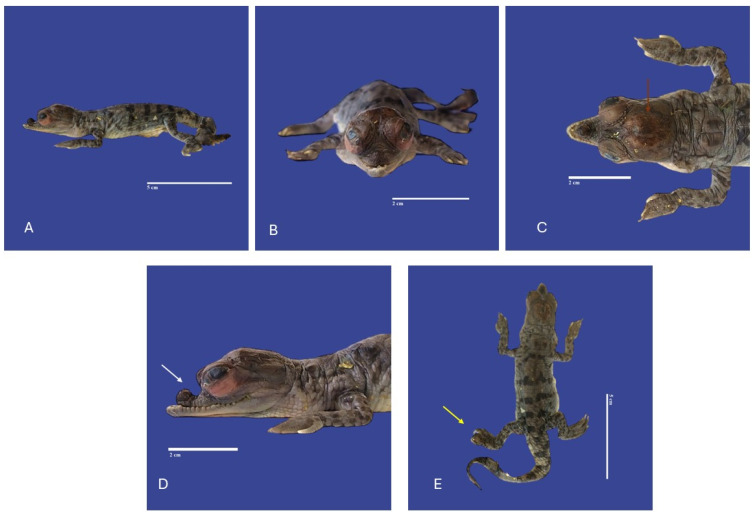
Kinked tail, maxillary micrognathia (↑ white), exophthalmos in both eyes, brachycephalic skull, ankylodactyly of the two digits fused in the left posterior member (↑ yellow), congenital cataract in both eyes, tubercles on top of the skull (↑ red), and epitheliogenesis imperfecta on the palpebra *Crocodylus acutus* embryo. ((**A**)—Lateral body view, (**B**)—Cranial view, (**C**)—ventrodoral head view, (**D**)—lateral head view, (**E**)—ventrodorsal view). Scale bar = 5 cm and 2 cm.

**Figure 14 vetsci-11-00317-f014:**
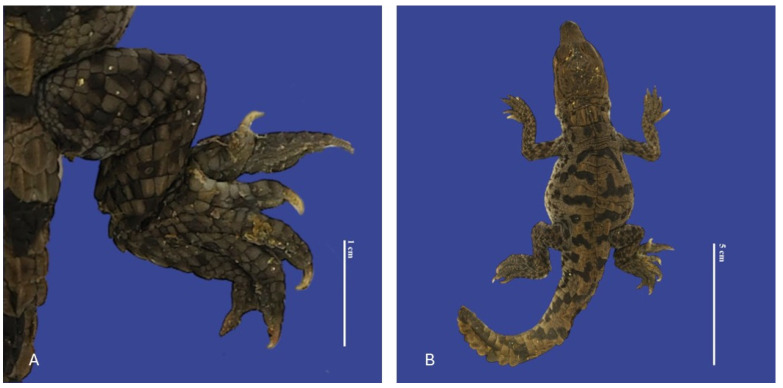
Polydactyly in the posterior right limb with two extra fingers, distended celomic cavity, tail blunt-tipped, bent tail, and brachyury in *Crocodylus acutus* embryo. ((**A**)—Limb view, (**B**)—ventrodorsal view). Scale bar = 1 cm and 5 cm.

**Figure 15 vetsci-11-00317-f015:**
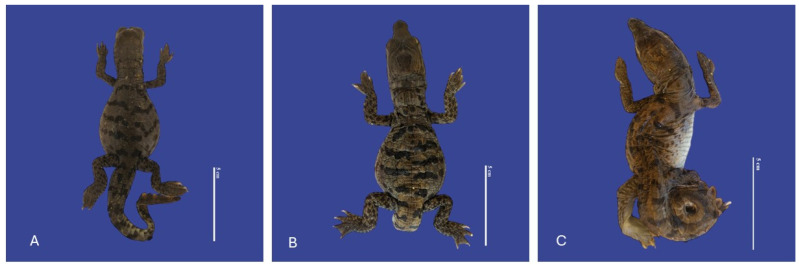
(**A**) Kinked tail; (**B**) acaudia; (**C**) curved tail *Crocodylus acutus* embryos. Scale bar = 5 cm.

**Figure 16 vetsci-11-00317-f016:**
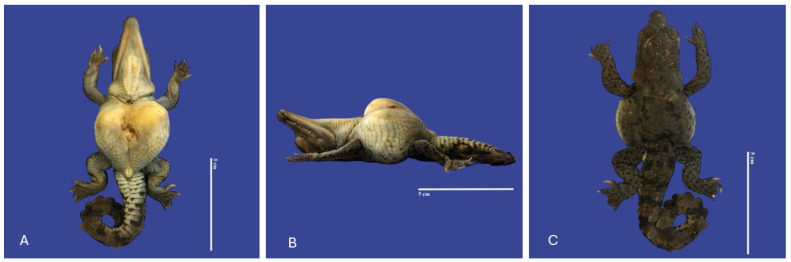
Achondroplasia, scoliosis, yolk sac retention, micromelia of the left upper member, and curly tail *Crocodylus acutus* embryo. ((**A**)—Ventrodorsally view, (**B**)—Lateral view, (**C**)—Dorsal ventral view). Scale bar = 5 cm.

**Figure 17 vetsci-11-00317-f017:**
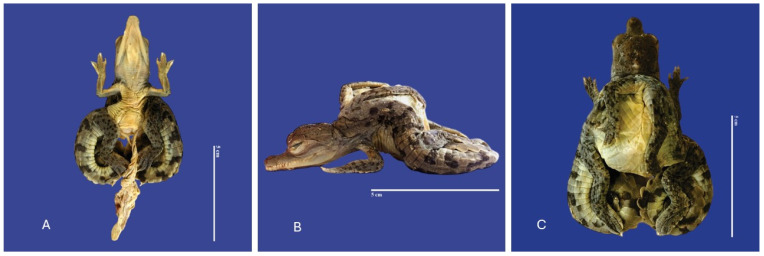
Cephalothoracopagus twins, curly tail, epitheliogenesis imperfecta at the top of the skull, and yolk sac retention in *Crocodylus acutus* embryo. ((**A**,**B**)—Lateral view, (**C**)—Ventrodorsal view). Scale bar = 5 cm.

**Figure 18 vetsci-11-00317-f018:**
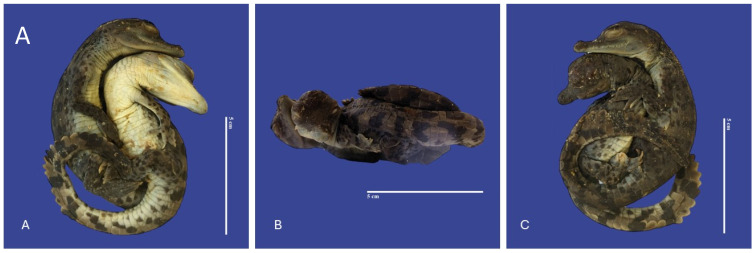
Monozygotic Symmetrical Twins of *Crocodylus acutus.* ((**A**)—Ventra dorsal view, (**B**)—Lateral view, (**C**)—Dorsal ventral view). Scale bar = 5 cm.

**Table 1 vetsci-11-00317-t001:** Information regarding nests (*n* = 61), lakes (*n* = 15), offspring, collection date, incubated eggs (*n* = 1242), laid eggs (*n* = 1773), broken eggs (*n* = 48), infertile eggs (*n* = 470), hatches (*n* = 705), and dead embryos (*n* = 505) from American crocodile (*Crocodylus acutus* Cuvier, 1807) from a farm in Colombia.

Individual Collected	Lake	Nest	Eggs Incubated	Eggs Broken	Total Eggs Laid	Infertile Eggs	Collection Date	Hatching Date	Hatches	Dead Embryos
N53	2	53	36	0	36	0	2 February 2023	25 April 23	20	16
N64	195	64	27	0	27	0	4 February 2023	25 April 23	13	14
N96A	9	96	17	0	17	16	8 February 2023	29 April 23	1	0
N96B	9	96	17	0	17	16	8 February 2023	29 April 23	1	0
N97	11	97	41	0	41	25	8 February 2023	2 May 2023	8	8
N105	8	105	17	0	17	3	9 February 2023	2 May 2023	1	13
N116	3	116	47	0	47	2	9 February 2023	25 April 23	33	12
N121	8	121	21	9	30	0	10 February 2023	5 May 2023	2	19
N156	15	156	23	1	24	4	12 February 2023	5 May 2023	4	15
N162	10	162	32	0	32	5	13 February 2023	2 May 2023	18	9
N167	2	167	48	0	48	1	13 February 2023	4 May 2023	28	19
N167G1	2	167	48	0	48	1	13 February 2023	4 May 2023	28	19
N167G2	2	167	48	0	48	1	13 February 2023	4 May 2023	28	19
N181A	10	181	39	0	39	0	14 February 2023	5 May 2023	19	20
N181B	10	181	39	0	39	0	14 February 2023	5 May 2023	19	20
N200	7	200	22	3	25	0	15 February 2023	5 May 2023	15	7
N202	7	202	19	0	19	0	15 February 2023	5 May 2023	15	4
N205	12	205	26	0	26	5	15 February 2023	4 May 2023	12	9
N206	11	206	33	0	33	29	15 February 2023	8 May 2023	2	2
N222	8	222	11	0	11	0	16 February 2023	7 May 2023	8	3
N241	7	241	29	0	29	0	17 February 2023	7 May 2023	18	11
N254	15	254	25	14	39	3	18 February 2023	4 May 2023	16	6
N264A	5	264	21	1	22	4	18 February 2023	11 May 2023	9	8
265B	9	265	30	0	30	2	18 February 2023	13 May 2023	15	13
N281	195	281	15	0	15	1	19 February 2023	10 May 2023	14	0
N283	195	283	30	0	30	5	19 February 2023	8 May 2023	14	11
N291	6	291	28	0	28	15	20 February 2023	11 May 2023	8	5
N292	5	292	32	0	32	32	27 March 2023	9 May 2023	0	0
N315	1	315	27	0	27	10	21 February 2023	9 May 2023	8	9
N333	9	333	23	0	23	5	22 February 2023	17 May 2023	6	12
N334	8	334	9	0	9	0	22 February 2023	13 May 2023	8	1
N352C	9	352	38	0	38	5	23 February 2023	15 May 2023	24	9
N366	11	366	38	2	40	37	31 March 2023	13 May 2023	0	1
N367	10	367	37	0	37	7	24 February 2023	17 May 2023	19	11
N368	10	368	35	0	35	1	24 February 2023	17 May 2023	25	9
N385	3	385	30	0	30	1	24 February 2023	17 May 2023	15	14
N402	16	402	7	0	7	7	1 April 2023	14 May 2023	0	0
N404	2	404	31	0	31	3	25 February 2023	17 May 2023	16	12
N404A	2	404	31	0	31	3	25 February 2023	17 May 2023	16	12
N405	3	405	30	0	30	28	25 February 2023	17 May 2023	1	1
N405A	3	405	30	0	30	28	25 February 2023	17 May 2023	1	1
N420	11	420	37	0	37	29	26 February 2023	17 May 2023	5	3
N442	7	422	36	0	36	1	27 February 2023	15 May 2023	35	0
N443	6	443	26	0	26	13	28 February 2023	17 May 2023	7	6
N453	1	453	30	0	30	29	1 March 2023	18 May 2023	0	1
N455	4	455	37	1	38	0	1 March 2023	23 May 2023	27	10
N457	8	457	26	0	26	0	1 March 2023	21 May 2023	22	4
N459A	6	459	20	0	20	0	1 March 2023	20 May 2023	17	3
N459	6	459	20	0	20	0	1 March 2023	20 May 2023	17	3
N486	7	486	30	0	30	1	3 March 2023	21 May 2023	24	5
N489	195	489	20	0	20	1	3 March 2023	28 May 2023	16	3
N492	12	492	23	0	23	5	3 March 2023	21 May 2023	8	10
N496	4	496	40	0	40	36	3 March 2023	20 May 2023	0	4
N502	2	502	19	0	19	19	5 March 2023	22 May 2023	0	0
N504A	4	504	33	2	35	10	5 March 2023	23 May 2023	15	8
N504	4	504	33	2	35	10	5 March 2023	23 May 2023	15	8
N506	8	506	26	2	28	1	5 March 2023	23 May 2023	11	14
N509	7	509	26	0	26	2	6 March 2023	23 May 2023	9	15
N509A	7	509	26	0	26	2	6 March 2023	23 May 2023	9	15
N531	4	531	15	11	26	0	8 March 2023	13 May 2023	5	10
N551	7	551	15	0	15	6	14 March 2023	31 May 2023	0	9

**Table 2 vetsci-11-00317-t002:** Malformations in American crocodile (*Crocodylus acutus*) from a farm in Colombia.

Body Region	Teratology	Number
Skin	Ephitheliogenesis imperfecta	5
	Depigmentation	3
	Leucism	2
Spine	Scoliosis	14
	Lordosis	3
	Kyphosis	2
	Campylorrachis scoliosa	1
Head	Anencephaly/prosencephalic hypoplasia	1
	Laterognathia	12
	Meningoencephalocete	3
	Brachygnatia	1
	Microcephaly	4
	Maxillary micrognathia	3
	External hydrocephalus	1
	Maxillary macrognathia	7
	Acrania	3
	Acephaslostomia	1
	Maxilla Agnathia	1
	Brachycephalic skull	1
Eye and nose	Congenital cataract	2
	Anophthalmia	1
	Exophthalmia	7
	Microphthalmia	1
	Atresia	2
Limbs	Micromelia	1
	Polydactyly	2
	Carpal flexure	1
	Ankylodactylia	2
	Malrotation	1
	Arthrogryposis	1
Tail	Curled tail	17
	Kinked tail	18
	Acaudia	16
	Brachyuria	3
Others	Achondroplasia (Dwarfism)	4
	Distended celomic cavity	1
	Ectopia cordis	1
	Yolk sac retention	47
	Gastroschisis	1
	Monozygotic symmetrical twins	
	Thoracopagus, sternopagus, xiphopagus twins	
	Cephalothoracopagus twins	

## Data Availability

Data are contained within the article.

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
