# Peer review of "Congenital Anomalies in American Crocodile (Crocodylus acutus, Cuvier, 1807) Embryos from a Farm Breeder in Colombia"

_vetsci, 2024, doi:10.3390/vetsci11070317_

Round 1

Reviewer 1 Report

Comments and Suggestions for Authors

The authors report the congenital defects detected in the embryos of the American crocodile (Crocodylus acutus) collected nearby a Colombian farm. The work is very interesting and the major strength is the good and detailed series of images about the malformations observed in different anatomical districts. However, for some heavy weakness about the samples numbers (deserving of clarification and deep check), use of scientific terms and way of presenting results, I suggest a major revision. Following my specific comments.

Line 27 and 35: limbs and tails malformations prevalence is 31.3% or 29%?

Line 37: not ziphopagus but (here and everywhere in the text) xiphopagus

Line 37: about acephalosomia, I’ve checked pub med and the web but I didn’t find this term; check the type of malformation and the correct definition.

Lines 83 and 99: the eggs considered in the study are 1465 or 1773?

Line 84 and Table 1: broken eggs are 36, or 48 as results by adding those listed in Table 1?

Line 84 and Table 1: infertile eggs are 421, or 470 as results by adding those listed in Table 1?

Line 84: after checking the correct total number of the eggs, I suggest to insert here (in accordance with Table 1) how many crocodiles, nests and lake the eggs came from.

Table 1: I suggest to insert a final line with the total number of each type of egg (eggs incubated, eggs broken, etc).

Line 99: it’s not clear how many embryos with malformations (among 550 eggs with dead embryos) have been observed and if the prevalence of 95.7% presenting more than one congenital defect is referred to 550 eggs with dead embryos or to 61 embryos included in the study. It seems that 4.3% (100 - 95.7) of embryos presented only one malformation or no malformation: is it correct?

Line 99: specify how and why the 61 embryos have been selected. A doubt: the total embryos showed in the Figures 2-17 are less (I think about 50) than 61 embryos included  (theoretically all these to be described and photographed if you have selected them among the total abnormal embryos).

Line 101 and Table 2: in Table 2 the listed types of anomalies are 42 and not 43 as in line 101.

Table 2: check carefully the terms of malformations (some are uncorrect or even not existing in scientific literature): ziphopagus/xiphopagus; leucotism?/leucism; prosenphalic hupoplasia/prosencephalic hypoplasia; external hydrocephaly/external hydrocephalus; acephalostomia?; atretic/better use atresia; micromei/micromelia; ankylodayly/ankylodactylia; malrotated/better use malrotation; arthogriposis/arthrogryposis;brachyuri/brachyuria; cephalothoracopagus (why here and not included among twins?); etopia cordis /ectopia cordis; retention yolk sac/better yolk sac retention.

Line 104-107: total number of malformation is 204 or 202 as results by adding those listed in Table 2?

Chapters 3.1 – 3.6: In this chapters (the core of the article) I suggest to give a brief description of various malformations (as for example you have correctly made for leucism) and not report only their number  which is a mere repetition of the data in Table 2.

Line 238: Probably “inspite of”.

Line 241: It would be interesting, if known, insert here the chemical products present in polluted farms’ water in order to compare for example with those in line 234-235 (observed by other Authors as cause of malformations in crocodiles).

Author Response

The authors report the congenital defects detected in the embryos of the American crocodile (Crocodylus acutus) collected nearby a Colombian farm. The work is very interesting and the major strength is the good and detailed series of images about the malformations observed in different anatomical districts. However, for some heavy weakness about the samples numbers (deserving of clarification and deep check), use of scientific terms and way of presenting results, I suggest a major revision. Following my specific comments.

Line 27 and 35: limbs and tails malformations prevalence is 31.3% or 29%?

Author answer: it is 29%, it was changed.

Line 37: not xiphopagus but (here and everywhere in the text) xiphopagus

Author answer:  it was changed all over the text.

Line 37: about acephalosomia, I’ve checked pub med and the web but I didn’t find this term; check the type of malformation and the correct definition.

Author answer:  it was a mistake in the writing the word was eliminated.

Lines 83 and 99: the eggs considered in the study are 1465 or 1773?

Author answer:  1773 eggs

Line 84 and Table 1: broken eggs are 36, or 48 as results by adding those listed in Table 1?

Author answer:  48 eggs

Line 84 and Table 1: infertile eggs are 421, or 470 as results by adding those listed in Table 1?

Author answer:  470 eggs

Line 84: after checking the correct total number of the eggs, I suggest to insert here (in accordance with Table 1) how many crocodiles, nests and lake the eggs came from.

Author Answer:  added

Table 1: I suggest inserting a final line with the total number of each type of egg (eggs incubated, eggs broken, etc).

Author Answer:  added

Line 99: it’s not clear how many embryos with malformations (among 550 eggs with dead embryos) have been observed and if the prevalence of 95.7% presenting more than one congenital defect is referred to 550 eggs with dead embryos or to 61 embryos included in the study. It seems that 4.3% (100 - 95.7) of embryos presented only one malformation or no malformation: is it correct?

Author Answer:  changed

Line 99: specify how and why the 61 embryos have been selected. A doubt: the total embryos showed in the Figures 2-17 are less (I think about 50) than 61 embryos included  (theoretically all these are to be described and photographed if you have selected them among the total abnormal embryos).

Author Answer:  from the correction before 61 embryons have malformation in the 500 analysed. The sentence was corrected to better understand. In the photos around 50 appear since they are only representative of the different malformations, such as tail deformities were reported and the authors think that is not a necessary tree or for images from the same anomalies.

Line 101 and Table 2: in Table 2 the listed types of anomalies are 42 and not 43 as in line 101.

Author Answer:  corrected to 42

Table 2: check carefully the terms of malformations (some are uncorrect or even not existing in scientific literature): xiphopagus/xiphopagus; leucotism?/leucism; prosenphalic hupoplasia/prosencephalic hypoplasia; external hydrocephaly/external hydrocephalus; acephalostomia?; atretic/better use atresia; micromei/micromelia; ankylodayly/ankylodactylia; malrotated/better use malrotation; arthogriposis/arthrogryposis;brachyuri/brachyuria; cephalothoracopagus (why here and not included among twins?); etopia cordis /ectopia cordis; retention yolk sac/better yolk sac retention.

Author Answer:  the authors are sorry for the mistakes, probably autocorrection from words. The terms have been corrected. Cephalothoracopagus was adde d to twins.

Line 104-107: total number of malformation is 204 or 202 as results by adding those listed in Table 2?

Author Answer: 202

Chapters 3.1 – 3.6: In this chapters (the core of the article) I suggest to give a brief description of various malformations (as for example you have correctly made for leucism) and not report only their number  which is a mere repetition of the data in Table 2.

Author Answer: we added the information in the results.

Line 238: Probably “inspite of”.

Author Answer: corrected

Line 241: It would be interesting, if known, insert here the chemical products present in polluted farms’ water in order to compare for example with those in line 234-235 (observed by other Authors as causes of malformations in crocodiles).

Author Answer: although the authors did not perform any sampling of the studies, information regarding compounds that have been found in other Colombian rivers has been added to the text.

Reviewer 2 Report

Comments and Suggestions for Authors

This manuscript presents very interesting data about congenital anomalies in American crocodile (Crocodylus 2 acutus, Cuvier, 1807) embryos. It's of great importance for maintaining wild species diversity. A few minor issues needs to be improved, particularly the presentation of the figures. 

1. For Fig 1, the reviewer asks the authors carefully check the map to avoid the political issues (if exist potentially).

2. The Classification "(Crocodylus acutus, Cuvier, 1807)" in Line 44 doesn't need to be repeated in the following text (including table title, figure legends), please keep the text concise.

3. In Line 82 "a total of 1465 eggs laid", but in Line 99 "the 1773 eggs laid", why??

4. Check the figures from Fig 1 to Fig 17, make them organized and beatutiful (clear scale bars, correct lable of A, B, C, D and corresponding to "A, B, C, D" in figure legends).

5. Because there are too many professional terms of teratology, the authors are requested to carefully check them for the accuracy. 

Author Response

This manuscript presents very interesting data about congenital anomalies in American crocodile (Crocodylus 2 acutus, Cuvier, 1807) embryos. It's of great importance for maintaining wild species diversity. A few minor issues needs to be improved, particularly the presentation of the figures.

  1. For Fig 1, the reviewer asks the authors carefully check the map to avoid the political issues (if exist potentially).

Author Answer: the map was checked.

  1. The Classification "(Crocodylus acutus, Cuvier, 1807)" in Line 44 doesn't need to be repeated in the following text (including table title, figure legends), please keep the text concise.

Author Answer: corrected

  1. In Line 82 "a total of 1465 eggs laid", but in Line 99 "the 1773 eggs laid", why??

Author Answer: it was a error, text was corrected

  1. Check the figures from Fig 1 to Fig 17, make them organized and beatutiful (clear scale bars, correct lable of A, B, C, D and corresponding to "A, B, C, D" in figure legends).

Author Answer: corrected

  1. Because there are too many professional terms of teratology, the authors are requested to carefully check them for the accuracy.

Author Answer: checked.

Reviewer 3 Report

Comments and Suggestions for Authors

ABSTRACT

            English must be revised. In line 34, it is “From the analyzed eggs…” is an example of error. Out of the that, the abstract is acceptable, it summarizes all the main findings.

INTRODUCTION

- Again, English has been an issue in the entire manuscript.

- Please, avoid using parenthesis for large elements. Insert then in the text using conjunctions, it will enrich the writing.

- In line 56 and 57, you rapidly mention the threats that this species face. I would lime more information about this topic, considering it is part of the backbone of your justification for this study.

- About the congenital defects section, I think it is very shallow of information. The authors bring interesting elements related to environmental factors that may cause these teratogenic defects. Please, bring more information, mainly about other reptiles that may be object of comparison. Another point is the justification for this study. You mention briefly in the ABSTRACT, but does not bring it up in the Introduction. Pure biological knowledge is not enough, an environmental justification is needed.

MATERIALS AND METHODS

- I clearly understand that these are scarce samples from a wild animal, but the authors could have considered performed some histological analysis after the gross documentation, unless these specimens go to be part of a museum exposition. If it is the case, tissue samples could have been taken and sequenced and compared to normal crocodiles to understand the altered genes related to each malformation and make the study more complete, but for a first report of malformations in this specie, I must say that the authors collected a great amount of varied congenital malformations.

RESULTS

            I have several suggestions to improve the disposal of your results:

- Regarding this sentence “In this study, only 61 embryos were included to represent the sample more, and no 103 repetitive anomalies were included. Regarding body regions: 4.9% (10/204) affected the 104 skin, 9.8% (20/204) the spine, 29% (61/204) the limbs and tail, and 25% (51/204) the head. 105 Other malformations were 27,9%.”, I would suggest a pie chart for a better visualization of the results.

- The photodocumentation is good, but the figures disposition is not. A black background would be much more aesthetic for the manuscript. Another suggestion is a zoom of the main region of each embryo to highlight the malformation, some arrows pointing important structures. This is very important because you only have gross anatomy as your results, so you have to explore more.

- Figures captions are very badly written. Please. Elaborate them better.

- The explanation of the malformations is very shallow and does not explore the nuances of malformation. Please, rethink this section because flaws in it would lead to rejection.

- Unfortunately, the authors did not explore possible intern malformation as in the heart, kidneys, lung and the other viscera.

DISCUSSION

- The first two paragraphs look like from a textbook chapter, not for a scientific article. Please, rethink then because every one knows the definition of teratology.

- Here is the opportunity to explore even more the environmental component related to congenital malformations. Please, add more information correlating types of environmental teratogens to other studies related to congenital malformations in reptile, mainly in crocodiles.

In summary, the study is shallow, but as the first of its kind, it is acceptable if the authors improve the writing and data disposition.

Comments on the Quality of English Language

Extensive editing of English language required

Author Response

ABSTRACT

            English must be revised. In line 34, it is “From the analyzed eggs…” is an example of error. Out of the that, the abstract is acceptable, it summarizes all the main findings.

Author Answer: corrected

INTRODUCTION

- Again, English has been an issue in the entire manuscript.

Author Answer: corrected

- Please, avoid using parenthesis for large elements. Insert then in the text using conjunctions, it will enrich the writing.

Author Answer: corrected

- In line 56 and 57, you rapidly mention the threats that this species face. I would lime more information about this topic, considering it is part of the backbone of your justification for this study.

Author Answer: more information was added

- About the congenital defects section, I think it is very shallow of information. The authors bring interesting elements related to environmental factors that may cause these teratogenic defects. Please, bring more information, mainly about other reptiles that may be object of comparison. Another point is the justification for this study. You mention briefly in the ABSTRACT, but does not bring it up in the Introduction. Pure biological knowledge is not enough, an environmental justification is needed.

Author Answer: more information was added

MATERIALS AND METHODS

- I clearly understand that these are scarce samples from a wild animal, but the authors could have considered performed some histological analysis after the gross documentation, unless these specimens go to be part of a museum exposition. If it is the case, tissue samples could have been taken and sequenced and compared to normal crocodiles to understand the altered genes related to each malformation and make the study more complete, but for a first report of malformations in this specie, I must say that the authors collected a great amount of varied congenital malformations.

Author Answer: unfortunately was not possible to collect samples since these specimens are going to be part of a museum exposition in Colombia.

RESULTS

            I have several suggestions to improve the disposal of your results:

- Regarding this sentence “In this study, only 61 embryos were included to represent the sample more, and no 103 repetitive anomalies were included. Regarding body regions: 4.9% (10/204) affected the 104 skin, 9.8% (20/204) the spine, 29% (61/204) the limbs and tail, and 25% (51/204) the head. 105 Other malformations were 27,9%.”, I would suggest a pie chart for a better visualization of the results.

Author Answer: chart added

- The photodocumentation is good, but the figures disposition is not. A black background would be much more aesthetic for the manuscript. Another suggestion is a zoom of the main region of each embryo to highlight the malformation, some arrows pointing important structures. This is very important because you only have gross anatomy as your results, so you have to explore more.

Author Answer: regarding the colour blue was chosen and not the black because with the black some details were mised, and with the blue they can been seem better. Arrows were added.

- Figures captions are very badly written. Please. Elaborate them better.

- The explanation of the malformations is very shallow and does not explore the nuances of malformation. Please, rethink this section because flaws in it would lead to rejection.

Author Answer: information added

- Unfortunately, the authors did not explore possible intern malformation as in the heart, kidneys, lung and the other viscera.

Author Answer: unfortunately was not possible to collect samples since these specimens are going to be part of a museum exposition in Colombia

DISCUSSION

- The first two paragraphs look like from a textbook chapter, not for a scientific article. Please, rethink then because every one knows the definition of teratology.

Author Answer: removed

- Here is the opportunity to explore even more the environmental component related to congenital malformations. Please, add more information correlating types of environmental teratogens to other studies related to congenital malformations in reptile, mainly in crocodiles.

Author Answer: added

In summary, the study is shallow, but as the first of its kind, it is acceptable if the authors improve the writing and data disposition.

Round 2

Reviewer 3 Report

Comments and Suggestions for Authors

The authors have addressed all my comments, so I recommend for publication.